# Toxoplasmosis in Captive Ring-Tailed Lemurs (*Lemur catta*)

**DOI:** 10.3390/pathogens11101142

**Published:** 2022-10-03

**Authors:** Guido Rocchigiani, Niccolò Fonti, Simona Nardoni, Paolo Cavicchio, Francesca Mancianti, Alessandro Poli

**Affiliations:** 1Department of Veterinary Anatomy, Physiology and Pathology, Institute of Infection, Veterinary and Ecological Sciences, Leahurst Campus, University of Liverpool, Chester High Road, Neston CH64 7TE, UK; 2Dipartimento di Scienze Veterinarie, Università di Pisa, Viale delle Piagge, 2-56124 Pisa, Italy; 3Giardino Zoologico di Pistoia, Via pieve a Celle 160a, 51100 Pistoia, Italy

**Keywords:** *Toxoplasma gondii*, lemurs, fatal toxoplasmosis, MAT, IHC, genotyping, zoo animals

## Abstract

*Toxoplasma gondii* is one of the most common protozoan parasites and is widely present in all warm-blooded animals. Although clinical disease is uncommon, some species, including ring-tailed lemurs (*Lemur catta*), have been found to develop acute and lethal toxoplasmosis. The aim of this study was to describe the pathologic, immunohistochemical, serological, and molecular findings of an outbreak of fatal toxoplasmosis in three captive ring-tailed lemurs in Central Italy in 2009. The animals died acutely within few days. The necropsy was immediately performed; necrotic lesions in the spleen, liver, and kidney, as well as interstitial pneumonia, were found histologically. All animals had high titers of anti–*T. gondii*-specific antibodies (1:1280 IgM and 1:640 IgG) according to a modified agglutination test (MAT) and immunohistochemistry showed scattered tachyzoites in the target organs. Diagnosis was confirmed by PCR and clonal type II was identified. In addition, the seven co-habiting lemurs were seronegative. This paper reports the first outbreak of acute disseminated toxoplasmosis in captive ring-tailed lemurs in Italy. These findings confirm the high susceptibility of this endangered species to toxoplasma infection, which may be considered a further threat to captive population viability.

## 1. Introduction

*Toxoplasma gondii* is a coccidian parasite, member of the phylum Apicomplexa. Apparently, it can be considered the world’s most successful parasite: nearly all warm-blooded vertebrate species worldwide can be infected, including humans [1,2].

The transmission of *T. gondii* through carnivorism, fecal–oral, transplacental, and other secondary routes, such as lactogenic and venereal, is one factor contributing to its global diffusion [3]. Only domestic and wild felids can excrete environmentally resistant oocysts in their feces, thereby representing the only definitive hosts and playing a central role in life cycle and food and water contamination. The environment, including marine waterways, is heavily contaminated with *T. gondii* oocysts [4].

Although infection may be common in many mammalian and avian species, clinical disease is rare [3,5]. Nonetheless, some animal species show a higher susceptibility to this parasite and develop fatal toxoplasmosis [6]. Lemurs, New World non-human primates (NWNHPs), and Australasian marsupials are highly susceptible to toxoplasmosis with frequent sudden death when infected [7,8].

Ring-tailed lemurs (*Lemur catta*) are primates belonging to the suborder Strepsirrhini. Like other *Lemur* spp., they are endemic to Madagascar, where they are the only native primate species present [9].

In recent years, the loss of their native environment has had a significant impact on the preservation of the species [10]. Thus, many animals are kept in captivity, either as pets or in zoological collections. Lemurs can interact with a wide range of exotic and native wildlife in these facilities [6]. Furthermore, allowing children to approach and feed captive zoo animals in so-called “petting zoos” may worsen pathogen transmission in both directions [11]. Concerns about public health and further biodiversity loss are raised by this scenario.

Close proximity to wild felid species and stress due to captivity are two of the main risk factors for *T. gondii* infection [6]. A recent retrospective pathology review by Denk et al. confirmed *T. gondii* as a significant causative agent of sporadic and epizootic mortalities in zoo animal species, including lemurs [12].

In Central Italy, a captive population of ring-tailed lemurs is housed in the Giardino Zoologico di Pistoia (GZP), located in Pistoia, Tuscany. This 75.000 m^2^ zoo is a component of the EAZA Ex situ Programme (EEP) for the breeding of threatened species and shelters over 500 animals belonging to endangered species.

The aim of the study was to describe the pathologic, immunohistochemical, serological, and molecular findings of an outbreak of acute, fatal toxoplasmosis in three captive ring-tailed lemurs in Central Italy.

## 2. Results

The results of serological, molecular, and pathological studies for each lemur included in the present study are summarized in Table 1.

### 2.1. Clinics and Gross Examination

The three lemurs died acutely within a few days of each other. They showed non-specific clinical signs such as anorexia, dehydration, lethargy, and debilitation. Treatment of the animals with dexamethasone (Dexadreson, MSD Animal Health srl, Milano, Italy), enrofloxacin (Baytril 5%, Bayer S.p.A., Milano, Italy), NaCl 0.9%, and glucose 5% (Baxter S.p.A. Rome, Italy) was unsuccessful. Their deaths occurred within three days of the treatment beginning. During the postmortem examination, a slight enlargement of the liver, spleen, and kidney due to passive congestion was observed. Small, scattered necrotic foci were detected in the liver and spleen, together with a mild generalized acute pneumonia.

### 2.2. Histopathology and Immunohistochemistry

#### 2.2.1. Histopathology

Microscopical investigations showed the presence of necro-hemorrhagic inflammatory foci in the spleen, kidney, liver, and lungs (Figure 1).

Moderate to severe multifocal necro-hemorrhagic splenitis was found in all three deceased lemurs. Moderate multifocal, random, necrotizing hepatitis was present in 2/3 lemurs. The central nervous system and the heart did not show any histological changes, while multifocal to coalescing interstitial pneumonia, hyperemia, and edema were found in the lungs. Lesions involving liver and lung tissue were found only in the two adult females. In the kidney, cortical congestion, hyperemia of the glomerular capillaries, and mild interstitial nephritis were evident in all three subjects.

#### 2.2.2. Immunohistochemistry

The results of the immunohistochemical studies are presented in Figure 2.

Scattered tachyzoites were found in the spleen, liver, lungs, and kidney of adult female #4429, and the spleen, liver, and lungs of adult female #4376. Tachyzoites were also found in the spleen and kidneys of the infant male lemur. No cysts or pseudocysts were observed in the lemurs investigated.

### 2.3. Serology

All of the deceased lemurs were seropositive with the same high titers (1:1280 IgM and 1:640 IgG), while the sera of the seven live animals housed in the zoo were negative (Table 1).

### 2.4. Molecular Analysis

*T. gondii* DNA was detected in the spleen tissue of all three subjects and in the livers of the two adults. Subsequent RFLP-PCR analysis showed a type II banding pattern on several loci in all deceased lemurs (Table 2).

## 3. Discussion

The present paper describes, for the first time in Italy, fatal toxoplasmosis in three captive ring-tailed lemurs, confirming *T. gondii* as a cause of death. These results show that lemurs from Central Italy can be exposed to *T. gondii*, as reported for other zoo and captive animals in Italy [13,14].

In captive lemurs, similar cases of acute lethal toxoplasmosis have been reported in Madagascar [15], Japan [16], the Netherlands [17], the USA [18,19], Spain [20], Germany [21], China [22] and the UK [12]. Most of the clinical and pathologic findings described in this paper were compatible with those previously reported in Strepsirrhini and NWNHPs [7,23,24]. The absence of pathognomonic clinical manifestations, non-specific or even absent symptoms, and acute death were outlined. In many cases, lemurs exhibited depression, anorexia, dyspnea, and lethargy [12,17,18,19,20]. Even though the small sample size does not allow for statistical analysis, this report is consistent with the findings of Denk et al., who found that females were more likely to be infected [12]. Hepatomegaly, splenomegaly, and occasionally nephromegaly, discoloration, and multifocal-to-disseminated whitish pinhead dots are the macroscopic lesions that are most frequently seen in autopsies. The reddening of the mesenteric lymph nodes and pneumonia are also commonly reported [16,17,18,21]. In the present study, histopathology highlighted necrotizing hepatitis, splenitis, and interstitial pneumonia. The kidneys were also involved, with a hyperemic appearance, similar to what was reported by Borst and van Knapen [17]. In contrast to the localized toxoplasmosis described by Juan-Sallés et al. [20], no cardiac or enteric lesions were observed.

The diagnosis of toxoplasmosis was confirmed by immunohistochemistry and PCR. Immunostaining of *T. gondii* confirmed the infection in the three deceased lemurs. Scattered tachyzoites in the spleen, lungs, liver, and kidney were observed, while no tissue cysts were seen. These findings are consistent with previous non-human primate reports [7,17] and most likely support the hypothesis of acute and sudden death.

The immunohistochemical results were also corroborated by molecular analysis, as *T. gondii* DNA was detected in all three subjects. Following the PCR-RFLP method, the genotype detected in this outbreak belonged to clonal type II in multiple loci. Type II strains are the most commonly found in people from North America, Africa, and Europe [3]. Since a high genetic diversity of strains has been described in acute toxoplasmosis in non-human primates [7], further molecular studies will be useful to better understand the epidemiology of *T. gondii* in these animals.

Unfortunately, only two molecular studies on lemurs have been conducted in the USA [19] and China [22]. In all of those lemurs, clonal type II was identified. However, only Yang et al. [22] performed multi-locus genotyping. In Spencer’s work, only the SAG2 locus was genotyped, thereby hampering extensive multi-locus genotyping [19].

The sera of the three deceased lemurs showed anti-*T. gondii* antibody titers of 1280 IgM and 640 IgG, higher than values reported previously in clinical toxoplasmosis [17,19]. In the study by Borst and van Knapen [17], only 1/3 of dead lemurs were found to be seropositive. This could be certainly attributed to the peracute nature of the disease, which could prevent the mounting of a detectable immune response [25] and the lower sensitivity to the latex agglutination test (LAT) compared to the MAT [3,26]. In addition, all cohabitant lemurs were serologically tested to evaluate exposure to the protozoa, but none of them were positive according to the same technique. In two previous studies, blood samples were collected from surviving subjects and serology was performed [17,18]: no seropositivity was reported, suggesting that the surviving healthy lemurs did not encounter the pathogen.

However, not all NWNHPs or lemurs that come into contact with the parasites develop a lethal disease, as some studies have shown that seropositive primates may not be in life-threatening conditions, although seroprevalence is usually low [27,28,29,30].

Although the reasons for the lemurs’ high susceptibility to toxoplasmosis are not fully understood, it may be related to the fact that they had little to no contact with felines in ancestral times. As NWNHPs, this may have prevented them from developing effective defenses against the parasite [31]. Lemurs diverged from other primates about 63 million years ago and since there are no native felids in Madagascar, lemurs most likely evolved for at least 40 million years without any exposure to *T. gondii* [32,33]. Moreover, the arboreal habits of these species could minimize their contact with the few cat stools present in wildlife reserves [34]. It seems reasonable that the absence of a parasite–host coevolution, as well as differences in immunologic mechanisms between Strepsirrhini and other primates [35,36], could have contributed to the lack of an efficient immune response against *T. gondii* in lemurs.

The facility’s domestic cat population was the most likely epidemiologic link in our outbreak, as in many others described in the literature [6]. The most likely source of infection was oocyst contamination of the environment, water, and food. Supporting this hypothesis, the deceased lemurs had been observed to be in direct contact with some cats wandering around the park, and were grooming the felines through the dividing net of their area.

For risk management in zoos, the application of appropriate distances between all sensitive primate species and possible sources of infection, such as felines and visitors, as well as disinfection of their food, is recommended [7]. After this outbreak, the lemur area was separated from the park paths with fences, which prevent direct contact with the outside, especially with stray cats.

The preservation of the ring-tailed lemur in captivity has become an essential tool for the conservation of the species. Due to the destruction of its natural habitat and merciless hunting, the ring-tailed lemur was included by the IUNC in the list of species at risk of extinction [37]. Given the high pathogenicity of *T. gondii* in these species, the present investigation and further studies could provide useful data to improve health management in zoological parks and our knowledge of toxoplasma epidemiology.

To the best of our knowledge, the investigation described in this report is the first to combine immunohistochemistry, serology, and multi-locus genotyping on ring-tailed lemurs. Additionally, to date, no serological studies on captivity-reared ring-tailed lemurs have ever been carried out in Italy.

## 4. Materials and Methods

### 4.1. Sample Collection

A total of ten ring-tailed lemurs from the GZP (Pistoia, Italy; 43.9295° N, 10.8656° E) were included in the study. Of these, three (one male unweaned infant and two adult females who were, respectively, 6 and 14 years old) who spontaneously deceased in November 2009 were immediately sent to the Department of Veterinary Science, University of Pisa (Pisa, Italy) for necroscopic examination. Blood samples from the cardiac cavities were used for serology, while representative specimens from the lungs, liver, spleen, heart, kidney, brain, and small intestine were fixed in 10% buffered formalin solution (pH 7.4), stored at room temperature, and then routinely processed for histopathological examination. Spleen and liver samples were also placed in sterile 50 mL polypropylene tubes and stored at −20 °C for molecular analysis.

Moreover, seven sera from the healthy ring-tailed lemurs housed in the zoo (four adult females and three adult males) were added to the research to conduct a serological analysis. Serum samples were collected for routine health monitoring in March 2010 and were employed to evaluate exposure to the parasite.

The lemurs were housed in an outside enclosure with plants and natural soil (98 m^2^), and heated indoor halls during the coldest days of the year (A: 10 m^2^ indoor facility; B: 20 m^2^ indoor facility). Large glass windows in the two indoor facilities allowed the lemurs to follow the natural 24 h daylight cycle. Foods consisting of chopped fresh fruits and vegetables supplemented with monkey pellets (with minerals and vitamins), as well as cold tea, were administered two times per day.

### 4.2. Histopathology and Immunohistochemistry

Fixed lung, liver, spleen, heart, kidney, central nervous system, and small intestine tissue specimens were routinely dehydrated and paraffin embedded. Five-micron tissue sections were stained with hematoxylin–eosin (H&E). Other sections were mounted on polarized Menzel–Gläser Superfrost plus slides (Thermo Fisher Scientific, Waltham, MA, USA) for immunohistochemistry (IHC) with a 1:200 diluted rabbit polyclonal anti-*T.gondii* antibody (Thermo Fisher Scientific, Fremont, CA, USA) [38]. Sections of experimentally infected mouse spleen were used as a positive control. As a negative control, the primary antibodies were omitted.

The sections were subjected to heat-induced epitope retrieval (HIER) in sodium citrate buffer (pH 6) and Peroxidase Blocking Solution (Dako REAL, Carpinteria, CA, USA). Afterwards, the primary antibody was incubated for 1 h at room temperature, followed by the addition of a secondary biotinylated goat polyvalent antibody (Thermo Fisher Scientific, Fremont, CA). Finally, DAB (ImPACT DAB Peroxidase Substrate Kit, Vector, Burlingame, CA, USA) was used as indicated by the manufacturer’s instructions. Before any steps, TBS–Tween washing solution was applied to the sections.

### 4.3. Serology

The blood samples were all centrifuged and the serum obtained was subjected to a modified agglutination test (MAT) to quantify anti-*T. gondii* antibodies. A Toxo Screen DA test (BioMérieux, Lyon, France) with formalin-treated *T. gondii* tachyzoites as the antigen, was used according to the manufacturer’s instructions. The cut-off was 1:10 and positive sera were end-titrated using 2-fold dilutions [3]. The screening dilutions were 1/10 and 1/320 to avoid prozone phenomenon.

### 4.4. Molecular Analysis

Spleen and liver tissue samples underwent PCR procedures. DNA was extracted using a Dneasy Blood & Tissue Kit (Qiagen GmbH, Hilden, Germany) from about 25 mg of tissues, following the manufacturer’s instructions and stored at −20 °C before being used.

Firstly, a Nested-PCR Protocol targeting the *T. gondii* B1 gene was performed as described by Jones et al. [39]. Positive samples showed 96 bps bands as the positive control included. The positive control was DNA extracted from *T. gondii* tachyzoites used for culture. The DNA-positive sample (i.e., the spleen) showing the thickest band was selected for genotyping. RFLP-PCR of 12 genetic markers (SAG1, 3-SAG2, 5-SAG2, SAG2 new, SAG3, BTUB, GRA6, C22. 8, C29-2, L358, PK1, and Apico) was conducted as described by Su et al. [40].

## 5. Conclusions

This paper reports the first outbreak of diffuse and fatal toxoplasmosis in captive ring-tailed lemurs in Italy. *T. gondii* clonal type II was identified as the cause of death by serology, immunohistochemistry, and RFLP-PCR. These findings confirm the high susceptibility of this species to the protozoa and the importance of control measures in facilities housing captive animals.

## Figures and Tables

**Figure 1 pathogens-11-01142-f001:**
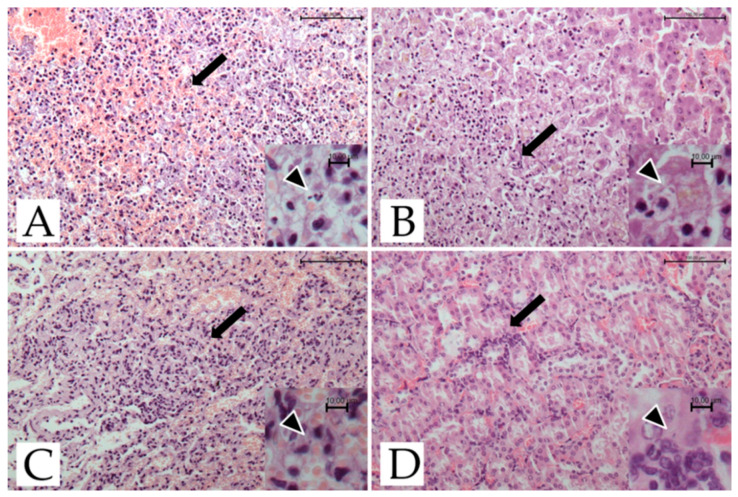
*T. gondii*-infected ring-tailed female adult lemur #4429, histological features. (**A**) Spleen: multifocal necrotizing and hemorrhagic splenitis (arrow; H-E; bar = 100 μm). Inset: scattered tachyzoites (arrowhead; H-E; bar = 10 μm). (**B**) Liver: focus of necrotizing hepatitis (arrow; H-E; bar = 100 μm). Inset: scattered tachyzoites (arrowhead; H-E; bar = 10 μm). (**C**) Lung: mild interstitial pneumonia with hyperemia (arrow; H-E; bar = 100 μm). Inset: scattered tachyzoites (arrowhead; H-E; bar = 10 μm). (**D**) Kidney: congestion of cortical area and mild interstitial nephritis (arrow; H-E; bar = 100 μm). Inset: scattered tachyzoites (arrowhead; H-E; bar = 10 μm).

**Figure 2 pathogens-11-01142-f002:**
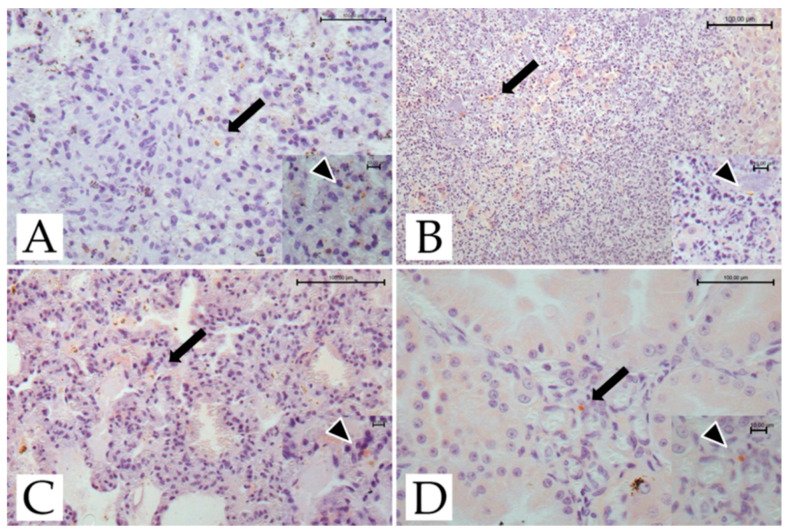
*T. gondii*-infected ring-tailed female adult lemur #4429, immunohistochemical findings. (**A**) Spleen: tachyzoites scattered throughout the organ (arrow; IHC, anti-toxoplasma; bar = 100 μm). Inset: scattered tachyzoites (arrowhead; IHC, anti-toxoplasma; bar = 10 μm). (**B**) Liver: tachyzoites scattered throughout the organ (arrow; IHC, anti-toxoplasma, bar = 100 μm). Inset: scattered tachyzoites (arrowhead; IHC, anti-toxoplasma; bar = 10 μm). (**C**) Lung: tachyzoites scattered in the pulmonary septa (arrow; IHC, anti-toxoplasma; bar = 100 μm). Inset: scattered tachyzoites (arrowhead; IHC, anti-toxoplasma; bar = 10 μm). (**D**) Kidney: tachyzoites scattered throughout the organ (arrow; IHC, anti- toxoplasma, bar = 100 μm). Inset: scattered tachyzoites (arrowhead; IHC, anti-toxoplasma; bar = 10 μm).

**Table 1 pathogens-11-01142-t001:** Health status, chip number, sex, age, and results of the ten ring-tailed lemurs included in the study.

Status	Chip	Sex	Age	Serum Titer (IgM/IgG)	PCR Positivity	Histologic Lesions	IHC Positivity
Acute death	n.d.*	M	infant	1:1280/1:640	Spleen	Spleen; kidneys	Spleen; kidneys
	4376	F	5y 7m	1:1280/1:640	Spleen; liver	Spleen; liver; lungs; kidneys	Spleen; liver; lungs
	4429	F	14y 5m	1:1280/1:640	Spleen; liver	Spleen; liver; lungs; kidneys	Spleen; liver; lungs; kidneys
Healthy	7860	F	17y 4m	<1:10	n.d.	n.d.	n.d.
	4169	M	21y 5m	<1:10	n.d.	n.d.	n.d.
	3388	M	19y 3m	<1:10	n.d.	n.d.	n.d.
	4541	F	16y 4m	<1:10	n.d.	n.d.	n.d.
	3040	F	17y 4m	<1:10	n.d.	n.d.	n.d.
	2498	M	12y 5m	<1:10	n.d.	n.d.	n.d.
	2877	F	11y 6m	<1:10	n.d.	n.d.	n.d.

* not determined.

**Table 2 pathogens-11-01142-t002:** The results of the genotyping assay for the three deceased PCR-positive lemurs.

Genetic Loci	Infant	Female #4376	Female #4429
SAG1 ^1^	II/III	II/III	II/III
3’SAG2	NA	II	II
5’SAG2 ^2^	I/II	I/II	I/II
SAG2 new	II	NA	NA
SAG3	II	II	II
BTUB	NA	II	II
GRA6	NA	NA	NA
C22-8	NA	NA	NA
C29-2	NA	II	II
L358	NA	II	II
PK1	NA	II	II
Apico	NA	NA	NA

^1^ At the SAG1 locus, genotypes II and III were undistinguishable; ^2^ At 5′SAG2, genotypes I and II were undistinguishable; NA = not amplified.

## Data Availability

Not applicable.

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
