# Peer review of "Toxoplasmosis in Captive Ring-Tailed Lemurs (*Lemur catta*)"

_pathogens, 2022, doi:10.3390/pathogens11101142_

Round 1
Reviewer 1 Report
The manuscript presents findings from a 2009 localized outbreak of Toxoplasma gondii with display ring tailed lemurs. The manuscript is well-organized and provides a thorough exploration of the incident.
That said, there are some improvements that could be made:
Line 164 "sentenced to death" is not the best word choice and suggest something along the line of "develop active disease".
Photomicrographs. In general, the photomicrographs are good, but could be improved by higher magnification of the IHC photomicrographs. The low magnification photos could be retained, but suggest either adding higher magnification or at least an inset with a higher magnification of the immunoreactive tachyzooites. For the HE photos, the lung photo is not as strong as it is described as it appears to have changes that look more along the lines of early autolysis and post-mortem dependency. I'd suggest a higher magnification photo that better characterizes the interstitial pneumonia.
Could you confirm that there were no gross findings? Does that reflect differences in who performed the necropsy versus who read out the slides? What was the time from death to necropsy?
Please define "SNC"
Has there been serologic monitoring of the population? Given that this incident occurred 13 years ago, it would be helpful to know about serosurveillance. I think if serum samples are available and funds are available, it would be of interest to pick some time periods and determine if there has been any seroconversion in the population.
Author Response
The manuscript presents findings from a 2009 localized outbreak of Toxoplasma gondii with display ring tailed lemurs. The manuscript is well-organized and provides a thorough exploration of the incident.
That said, there are some improvements that could be made:
- Line 164 "sentenced to death" is not the best word choice and suggest something along the line of "develop active disease".
As suggested by the editor, we have rephrased the sentence as “However, not all NWNHP or lemurs that come into contact with the parasites develop a lethal disease…”
- Photomicrographs. In general, the photomicrographs are good, but could be improved by higher magnification of the IHC photomicrographs. The low magnification photos could be retained, but suggest either adding higher magnification or at least an inset with a higher magnification of the immunoreactive tachyzooites. For the HE photos, the lung photo is not as strong as it is described as it appears to have changes that look more along the lines of early autolysis and post-mortem dependency. I'd suggest a higher magnification photo that better characterizes the interstitial pneumonia.
The photomicrograph quality has been drastically improved. The microscopic fields have been replaced using others in which the lesions were more evident, furthermore inserts have been added to highlight tachyzoites presence. To make the figures clearer, arrows have been added to evidence the lesions and arrowheads to highlight parasites in the inserts.
- Could you confirm that there were no gross findings? Does that reflect differences in who performed the necropsy versus who read out the slides? What was the time from death to necropsy?
The description of the macroscopic lesions in the text has been modified in order to highlight the lesions present, even if mild. The text has been modified by verifying what was highlighted by the different pathologists. To performe the necropsy, the subjects were transferred from the zoo to the Department where the necropsy room is located, this allowed to carried out the necropsy after 12/24 hours.
- Please define "SNC"
As suggested, we have replaced the acronym with “central nervous system” throughout the entire text.
- Has there been serologic monitoring of the population? Given that this incident occurred 13 years ago, it would be helpful to know about serosurveillance. I think if serum samples are available and funds are available, it would be of interest to pick some time periods and determine if there has been any seroconversion in the population.
We appreciate the reviewer’s insightful suggestion and agree that it would be useful to conduct further T. gondii serologic surveys on captive lemurs. However, given that the lemurs housed in the zoo showed no more clinical signs attributable to toxoplasmosis, and the serum samples analyzed in the current study were available due to routine monitoring following the outbreak and not regularly available, we determined that there was no need to subject the animals to additional stress by conducting additional serum sampling. Nonetheless, we are aware that, whilst also rarely reported, there are bibliographic references to asymptomatic seropositive subjects in the literature [30]. Thus, we believe that future aims could surely include investigations into possible T. gondii seroconversion not only in ring-tailed lemurs but also in other captive animals from the same aforementioned facility.
Reviewer 2 Report
Comments to the Authors:
This manuscript is a description of an outbreak of acute fatal toxoplasmosis in a captive lemur colony in Italy. The authors are particularly interested in establishing positive proof of the diagnosis of toxoplasmosis and present compelling data to that end. The authors are to be congratulated for clearly stating the aim of the paper. However, the rationale is not clearly articulated. Some problems with the manuscript arise in the presentation of the results. There are only minor problems with English language and grammar.
Abstract
Line 22-23: This sentence could be extended to express what should be the take home message: ring-tailed lemurs are an endangered species in the wild and could be equally endangered in captivity due to Toxoplasma infection. Isn’t this the true importance of your findings?
Introduction
Lines 38 – 41. It is not clear why NWNHP and OWNHP are mentioned at all since lemurs are neither of these. It would seem better to delete these sentences and then replace them with the description of ring-tailed lemurs, lines 44 – 46.
Line 47: provide a reference for this broad claim.
Results
Section 2.1:
How long after the onset of clinical signs before death? What is meant by “gross pathological changes”? Readers are not necessarily expert in the language of pathology.
Section 2.2.1 and Figure 1.
Figure 1: The magnification of all four panels is too low. The arrows in panel B do not seem to point to anything remarkably different from the rest of the tissue. Add an inset showing a much higher magnification; use the arrows or arrowheads in this inset. All four panels need arrowheads to point to the specific lesions or conditions stated in the legend. Please show a photomicrograph of normal tissue for comparisons.
Line 76: Which of the 3 lemurs did not have necrotizing hepatitis? Please explain “SNC”.
Line 82: Shouldn’t this be 2.2.2?
Figure 2: The magnification is too low to show any proof that there are tachyzoites present. The bar is 100 µm while a tachyzoite is 3 – 7 µm. There is no photographic proof of tachyzoites at this mag. The same comments from Figure 1 apply to this figure as well, including showing the no-primary control.
Section 2.3
This section could be enhanced with more details of the test results. Both of the tests listed in Materials and Methods give quantitative results. The text gives actual titres but the table is qualitative. Perhaps the magnitude of the anti-toxoplasma antibody response is a key marker for increased risk of death? Did all 3 deceased lemurs have the exact same titres?
Section 2.4
This section needs more detail. There are only 3 spleens and 2 livers. The authors could show the data for all these samples. The parasite load according to the pcr should be reported if possible.
The reference to Table 1 is misplaced. Since it shows demographics, serology, histology, and pcr results, it should be referenced at the beginning of Results section.
Table 2 footnotes: #2. This states that genotype I and III were undistinguishable. However, the table shows genotype II.
Discussion
Lines 146-149: Where, geographically, were these studies performed. Who is Spencer?
Line 161: The n=10 is too low to make this generalized comment. Additionally, the logic of this sentence is unclear.
Lines 187- 192: This is the real importance of the authors’ findings and should be emphasized much earlier in the manuscript.
Materials and Methods
Line 247: No reference number is given for Jones. Presumably it is #25.
Line 252: Reference #24 occurs after reference 25.
References:
Reference # 23 does not seem to appear in the text.
References 25 and 24 are out of order.
Author Response
Comments to the Authors:
This manuscript is a description of an outbreak of acute fatal toxoplasmosis in a captive lemur colony in Italy. The authors are particularly interested in establishing positive proof of the diagnosis of toxoplasmosis and present compelling data to that end. The authors are to be congratulated for clearly stating the aim of the paper. However, the rationale is not clearly articulated. Some problems with the manuscript arise in the presentation of the results. There are only minor problems with English language and grammar.
Abstract
- Line 22-23: This sentence could be extended to express what should be the take home message: ring-tailed lemurs are an endangered species in the wild and could be equally endangered in captivity due to Toxoplasma infection. Isn’t this the true importance of your findings?
We thank the reviewer for pointing this out. We have rephrased the sentence: “These findings confirm the high susceptibility of this endangered species to toxoplasma infection, which may be considered as a further threat to captive population viability.”
Introduction
- Lines 38 – 41. It is not clear why NWNHP and OWNHP are mentioned at all since lemurs are neither of these. It would seem better to delete these sentences and then replace them with the description of ring-tailed lemurs, lines 44 – 46.
We agree with the reviewer and, as suggested, the sentence has been deleted and replaced with “Lemurs, New World non-human primates (NWNHP), and Australasian marsupials are highly susceptible to toxoplasmosis with frequent sudden death when infected [7,8]. Ring-tailed lemurs (Lemur catta) are primates belonging to the suborder Strepsirrhini. Like other Lemur spp., they are endemic to Madagascar, where they are the only native primate species present [9].”
- Line 47: provide a reference for this broad claim.
Reference #10 (https://doi.org/10.1002/ece3.6337) was added as suggested.
Results
- Section 2.1.
How long after the onset of clinical signs before death? What is meant by “gross pathological changes”? Readers are not necessarily expert in the language of pathology.
The data required have been added in the text, lines 90-94. The three lemurs died acutely within a few days of each other. The animals lived in a group in two indoor facilities and showed aspecific clinical signs as reported in the manuscript. The description of gross pathological changes has been properly modified.
- Section 2.2.1 and Figure 1.
Figure 1: The magnification of all four panels is too low. The arrows in panel B do not seem to point to anything remarkably different from the rest of the tissue. Add an inset showing a much higher magnification; use the arrows or arrowheads in this inset. All four panels need arrowheads to point to the specific lesions or conditions stated in the legend. Please show a photomicrograph of normal tissue for comparisons.
The photomicrograph quality has been drastically improved. The microscopic fields have been replaced using others in which the lesions were more evident, furthermore inserts have been added to highlight tachyzoites presence. To make the figures clearer, arrows have been added to evidence the lesions and arrowheads to highlight parasites in the inserts. We think that the new figures should clearly highlight the lesions described. The addition of a healthy tissue would excessively complicate the table of figures with a reduction of the photomicrographs.
- Line 76: Which of the 3 lemurs did not have necrotizing hepatitis? Please explain “SNC”.
In section 2.2.1. we have written: “Lesions involving liver and lung tissue were found only in the two adult females”. Besides, we have replaced as suggested the acronym SNC with “central nervous system” throughout the entire text.
- Line 82: Shouldn’t this be 2.2.2?
Typo was changed a suggested
- Figure 2: The magnification is too low to show any proof that there are tachyzoites present. The bar is 100 µm while a tachyzoite is 3 – 7 µm. There is no photographic proof of tachyzoites at this mag. The same comments from Figure 1 apply to this figure as well, including showing the no-primary control.
Also Figure 2 has been drastically modified and inserts have been added to highlight the parasite presence. We think that the new figures should clearly highlight the lesions described. The addition of a negative control would excessively complicate the table of figures with a reduction of the photomicrographs.
- Section 2.3
This section could be enhanced with more details of the test results. Both of the tests listed in Materials and Methods give quantitative results. The text gives actual titres but the table is qualitative. Perhaps the magnitude of the anti-toxoplasma antibody response is a key marker for increased risk of death? Did all 3 deceased lemurs have the exact same titres?
We thank the reviewer for the suggestions: we have employed quantitative methods and all three deceased lemurs scored the same results. Since the next 2-fold dilutions were negative (1:2560 and 1:1280), we can only conclude that the titer was in this range, without asserting whether the three animals had the same exact serum concentration. Nevertheless, we agree that it could be explained better, so we rephrased it in “All the deceased lemurs were seropositive with the same high titre (1:1280 IgM and 1:640 IgG), while the sera of the seven live animals housed in the zoo scored negative (< 1.10 as reported in Table 1).
Concerning the use of the antibody titer as a key marker, we believe that the high inter-individual variability reported in the literature may make serology difficult to be used as a prognostic factor. In fact, even though the clinical and pathological picture in our work was similar to that of Spencer and Borst and Van Knapen [17,19], the serology results were not. Certainly, future research would be beneficial in clarifying this issue. Finally, as suggested, we have added quantitative data in Table 1 and moved it to the beginning of the Results section.
- Section 2.4
This section needs more detail. There are only 3 spleens and 2 livers. The authors could show the data for all these samples. The parasite load according to the pcr should be reported if possible.
Thanks for the suggestion. We appreciate that this part is relatively brief but since PCR and genotyping of T. gondii strains are extensively treated elsewhere (Su et al., 2010; [40]), we deem that such data are lengthy and not necessary here. We consider that a table with all loci is much more valuable and immediate than any gel picture. Unless you specifically request it, we would prefer not to include such data in the manuscript. The parasite load was not assessable as we did an end-point PCR.
- The reference to Table 1 is misplaced. Since it shows demographics, serology, histology, and pcr results, it should be referenced at the beginning of Results section.
Thanks for the suggestion. Done as suggested.
- Table 2 footnotes: #2. This states that genotype I and III were undistinguishable. However, the table shows genotype II.
Thanks for the suggestion. We removed it, as suggested.
Discussion
- Lines 146-149: Where, geographically, were these studies performed. Who is Spencer?
Information and reference were added to the text.
- Line 161: The n=10 is too low to make this generalized comment. Additionally, the logic of this sentence is unclear.
As suggested by the author the generalized comment was deleted.
- Lines 187- 192: This is the real importance of the authors’ findings and should be emphasized much earlier in the manuscript.
As suggested by the reviewer we have stressed more this issue both in the abstract and in the introduction (lines 80-88).
Materials and Methods
- Line 247: No reference number is given for Jones. Presumably it is #25.
We thank the reviewer for pointing out this typo. We have fixed it by adding the reference number [39].
- Line 252: Reference #24 occurs after reference 25.
The references’ order has been checked and corrected as suggested.
References:
- Reference # 23 does not seem to appear in the text. References 25 and 24 are out of order.
The references’ order has been checked and corrected as suggested.
Round 2
Reviewer 2 Report
Thank you for the extensive text modifications, enhancement of the photomicrographs, and reasonable, thoughtful responses. While this reviewer does not agree with some of the responses, they (the responses) are nonetheless acceptable. The paper is greatly improved. There are just a few minor English grammar edits (for example, Line 64 "lemurs" should be "lemur" ) that should be corrected by copy editors.